# Evaluation of Possible Neobavaisoflavone Chemosensitizing Properties towards Doxorubicin and Etoposide in SW1783 Anaplastic Astrocytoma Cells

**DOI:** 10.3390/cells12040593

**Published:** 2023-02-12

**Authors:** Mateusz Maszczyk, Klaudia Banach, Jakub Rok, Zuzanna Rzepka, Artur Beberok, Dorota Wrześniok

**Affiliations:** Department of Pharmaceutical Chemistry, Faculty of Pharmaceutical Sciences in Sosnowiec, Medical University of Silesia in Katowice, 41-200 Sosnowiec, Poland

**Keywords:** apoptosis, cytostatic agents, doxorubicin, etoposide, glioma, isoflavones, flavonoids

## Abstract

Flavonoids exert many beneficial properties, such as anticancer activity. They were found to have chemopreventive effects hindering carcinogenesis, and also being able to affect processes important for cancer cell pathophysiology inhibiting its growth or promoting cell death. There are also reports on the chemosensitizing properties of flavonoids, which indicate that they could be used as a support of anticancer therapy. It gives promise for a novel therapeutic approach in tumors characterized by ineffective treatment, such as high-grade gliomas. The research was conducted on the in vitro culture of human SW1783 anaplastic astrocytoma cells incubated with neobavaisoflavone (NEO), doxorubicin, etoposide, and their combinations with NEO. The analyses involved the WST-1 cell viability assay and image cytometry techniques including cell count assay, Annexin V assay, the evaluation of mitochondrial membrane potential, and the cell-cycle phase distribution. We found that NEO affects the activity of doxorubicin and etoposide by reducing the viability of SW1783 cells. The combination of NEO and etoposide caused an increase in the apoptotic and low mitochondrial membrane potential subpopulations of SW1783 cells. Changes in the cell cycle were observed in all combined treatments. These findings indicate a potential chemosensitizing effect exerted by NEO.

## 1. Introduction

Flavonoids are bioactive phenolic compounds of natural origin, widely present in plant-derived products such as fruits, vegetables, and beverages [1]. Depending on the modifications of a flavonoid’s basic structure, such as degree of oxidation and unsaturation, they can be subdivided into anthocyanidins, chalcones, flavanols (catechins), flavanones, flavonols, flavones, and isoflavones [1,2]. The intake of these compounds in the diet has been linked with several positive health effects, which include i.a. diabetes prevention, reduction of the metabolic syndrome risk, neuroprotection, and a decrease in the chance of developing cardiovascular disease [3,4,5]. A substantial part of the research focused on flavonoids is also concentrated on their anticancer effects. Several of these substances have been found to lower the risk of such cancers as colorectal (flavonols, flavones, and anthocyanidins) [6], breast (flavonols and flavones) [7,8], or prostate (isoflavones and catechins) [9,10,11]. Hence, flavonoids are often referred to as chemopreventive agents, which are assumed to hinder carcinogenesis at its early stage [12,13]. Anticancer effects of flavonoids are also connected with their ability to affect the pathological processes of neoplastic cells such as abnormal growth or avoidance of programmed cell death—many of these substances have anti-proliferative and anti-metastatic properties and are able to promote apoptosis [1,14,15]. Among others, they were found to interfere with signaling pathways important for the pathophysiology of cancer cells such as AKT/mTOR or Ras/ERK, activate pro-apoptotic proteins, and also impede the cell cycle through inhibition of specific cyclins and cyclin-dependent kinases (CDKs) [15]. The pleiotropic molecular action of flavonoids indicates their potential to support anticancer therapy [16]. Furthermore, these substances were shown to sensitize tumor cells towards antineoplastic agents or even overcome chemoresistance [17,18,19,20,21,22,23]. For this reason, flavonoids are one of the subjects of scientific studies seeking a solution to the still unresolved problem of ineffective cancer treatment.

One of the types of tumors characterized by poor response to treatment is high-grade gliomas, classified by the World Health Organization (WHO) as grades III and IV, such as anaplastic astrocytoma (grade III) or glioblastoma (grade IV) [24,25]. Low effectiveness of the current therapeutical strategy including surgical resection, after which patients undergo radiotherapy and chemotherapy with temozolomide, is one of the main reasons for high mortality—it improves overall survival of high-grade glioma patients by only a few months [24,26,27]. Prime causes of this are a fast progression of the tumor and the resistance to temozolomide caused by expression of O^6^-methylguanine-DNA methyltransferase (MGMT) reversing the changes in DNA caused by the drug, which occurs in most cases of high-grade gliomas [28,29,30]. Despite continuous advances in the development of novel therapeutic options, such as immunotherapy or gene therapy, so far no solution that could give significant improvements was found [31,32]. Chemotherapeutics that are used in the treatment of other neoplasms are also being considered as a potential alternative to the current regimen of high-grade gliomas. Among them are such drugs as etoposide or doxorubicin and topoisomerase inhibitors, the satisfactory effectiveness of which has been demonstrated in several studies regarding anaplastic astrocytoma [33,34,35] and glioblastoma [35,36,37] in monotherapy regimens or in combination with other drugs. However, the low bioavailability to the central nervous system of these compounds and the possibility of chemoresistance development challenge their use in clinical practice [38,39]. Therefore, a combined treatment with a substance that could sensitize tumor cells to the chemotherapeutic to improve efficacy seems to be noteworthy [40,41,42].

Previously, we showed that the flavonoid of *Psoralea corylifolia* (L.), neobavaisoflavone (NEO), affects the in vitro activity of doxorubicin and etoposide in human glioblastoma U-87 MG cells by apoptosis enhancement and cell cycle alterations [43]. We also found that this isoflavone increases the effect of doxorubicin on the viability of U-87 MG cells [44]. It prompted us to investigate further the possible chemosensitizing properties of NEO on another high-grade glioma cell line. Herein, the aim of the presented work was to explore the influence of NEO in combination with doxorubicin or etoposide on the viability and growth of human SW1783 anaplastic astrocytoma cells, as well as apoptosis, mitochondrial membrane potential, and the cell cycle.

## 2. Materials and Methods

### 2.1. Reagents and Chemicals

Human anaplastic astrocytoma cells (SW1783, HTB-13^™^) were obtained from ATCC (Manassas, VA, USA). Neobavaisoflavone (7-hydroxy-3-(4-hydroxy-3-(3-methyl-2-buten-1-yl)phenyl)-4H-1-benzopyran-4-one), penicillin G, and dimethyl sulfoxide (DMSO) were retrieved from Sigma-Aldrich Inc. (St. Louis, MO, USA). Dulbecco’s modified Eagle’s medium (DMEM) and fetal bovine serum (FBS) were acquired from Cytogen (Zgierz, Poland). Trypsin/EDTA solution was purchased from ThermoFisher Scientific (Waltham, MA, USA). Cell proliferation reagent WST-1 was obtained from Roche (Mannheim, Germany). NC-Slides™ A8 and Via1-Cassettes™, as well as Solution 3 (1 μg/mL DAPI, 0.1% Triton X-100 in PBS), Solution 7 (200 μg/mL JC-1 in DMSO), Solution 8 (1 μg/mL DAPI in PBS), Solution 15 (500 μg/mL Hoechst 33342, aqueous), Solution 16 (500 μg/mL propidium iodide, aqueous), were retrieved from ChemoMetec (Lillerød, Denmark). Neomycin sulfate was acquired from Amara (Kraków, Poland). Annexin V binding buffer and Annexin V-CF488A conjugate were obtained from Biotium (Fremont, CA, USA). In the study, the following drugs were used: doxorubicin (Doxorubicin Accord, Accord, Ahmedabad, India), etoposide (Etoposid-Ebewe, Ebewe Pharma, Ahmedabad, India), irinotecan (Irinotecan Accord, Accord, Ahmedabad, India). The rest of the chemicals were purchased from POCH S.A. (Gliwice, Poland).

### 2.2. Cell Culture

The research was performed on the human anaplastic astrocytoma SW1783 cell line, which was obtained from ATCC (HTB-13^™^, Manassas, VA, USA). Cells were cultured in DMEM supplemented with FBS (final concentration 10%), penicillin G (10,000 U/mL), neomycin (10 μg/mL), and amphotericin B (0.25 mg/mL) at 37 °C and humidified 5% CO_2_ atmosphere.

Prior to the experiment, SW1783 cells were seeded in 96-well microplates (3000 cells/well) and T-75 flasks (1 × 10^6^ cells/flask), and preincubated for 48 h (37 °C, 5% CO_2_). Afterward, the medium was removed and the drug solutions, prepared in DMEM, were added. Following 48 h of treatment, cells cultured in flasks were detached by trypsinization, centrifuged, and resuspended in the medium for further analyses.

### 2.3. Cell Viability Assay

The viability of the cells was estimated using a WST-1 (4-[3-(4-iodophenyl)-2-(4-nitrophenyl)-2H-5-tetrazolio]-1,3-benzene disulfonate) colorimetric assay. This analysis is based on the action of mitochondrial dehydrogenases that catalyze the reduction of the WST-1 reagent. The amount of the product correlates with the number of metabolically active cells. Human anaplastic astrocytoma SW1783 cells were seeded at 3000 cells per well in 96-well microplates in a supplemented DMEM growth medium and incubated for 48 h at 37 °C and 5% CO_2_. Then, the medium was replaced with solutions of NEO (1–75 μM), doxorubicin (1–50 μM), etoposide (1–50 μM), irinotecan (1–50 μM), and NEO (25 μM, 75 μM) combined with doxorubicin (1 μM), etoposide (10 μM), and irinotecan (10 μM). After 48 h of incubation, 10 μL of WST-1 reagent was added to each well and after 1h of incubation, the absorbance of the samples was measured at 440 nm and, as a reference wavelength, 650 nm using an Infinite 200 PRO (TECAN, Männedorf, Switzerland) microplate reader. The controls were normalized to 100% for each assay, and the results were shown as the percentage of the controls.

### 2.4. Cell Count Assay

The total number of cells was estimated using NucleoCounter^®^ NC-3000^™^ fluorescence image cytometer controlled by NucleoView NC-3000 Software 2.1.25.12 (Chemometec, Denmark). After the 48 h of incubation with NEO (25 μM), doxorubicin (1 μM), doxorubicin-NEO mix (1 μM + 25 μM), etoposide (10 μM), and etoposide-NEO mix (10 μM + 25 μM) cells were trypsinized and loaded into Via1-Cassette™ (ChemoMetec) containing fluorescent dyes: DAPI (4′,6′-diamidino-2-phenylindole) staining dead cells and acridine orange staining the whole cell population.

### 2.5. Detection of Apoptotic Cells

The Annexin V assay was conducted to detect programmed cell death. One of the characteristics of apoptosis is the phosphatidylserine translocation to the outer surface of the cell membrane. In this analysis, fluorescent Annexin V conjugate that specifically binds to phosphatidylserine was utilized to reveal apoptotic cells. In brief, 4 × 10^5^ of cells in each sample were suspended in 100 μL of Annexin V binding buffer containing 2 μL of Annexin V-CF488A conjugate and 2 μL of Solution 15 (Hoechst 3334, 500 μg/mL) and incubated for 15 min at 37 °C. Then, stained cells were centrifuged at 400× *g* for 5 min and washed twice with Annexin V binding buffer. After the supernatant had been discarded, cells were resuspended in 100 μL of Annexin V binding buffer and stained with 2 μL of Solution 16 (propidium iodide, 500 μg/mL). The samples were loaded into NC-Slides^™^ A2 and analyzed immediately with NucleoCounter^®^ NC-3000^™^ fluorescence image cytometer.

### 2.6. Mitochondrial Membrane Potential Assay

The mitochondrial membrane potential (∆Ψm) was assessed with NucleoCounter^®^ NC-3000^™^ fluorescence image cytometer using JC-1 (5,5′,6,6′-tetrachloro-1,1′,3,3′-tetraehtylbenzimidaziolocarbocyanine iodide) stain. The analysis is based on the ability of the dye to accumulate inside healthy mitochondria, characterized by high ∆Ψm status, in its polymerized form emitting red fluorescence. In turn, in unhealthy cells where the mitochondria have low ∆Ψm, JC-1 localizes in the cytoplasm in its monomeric form fluorescing green. Following the treatment, 1 × 10^6^ cells were suspended in 12.5 μL of Solution 7 (JC-1, 200 μg/mL) and incubated at 37 °C. After 15 min, samples were centrifuged (400× *g* for 5 min) and washed twice with PBS. The cell pellets were resuspended in 250 μL of Solution 8 (DAPI in PBS, 1 μg/mL), loaded into NC-Slides^™^ A8, and analyzed immediately with the image cytometer.

### 2.7. Analysis of Cell Cycle

The cell cycle was analyzed using the NucleoCounter^®^ NC-3000^™^ fluorescence image cytometer. The principle of the method is based on the measurement of DNA content within cells. After the treatment, the obtained cell pellets were resuspended in 500 μL of PBS and fixed with 4.5 mL of 70% cold ethanol for at least 12 h and kept at 0–4 °C. Next, cells were stained with Solution 3 (DAPI and 0.1% Triton X-100 in PBS, 1 μg/mL) for 5 min (at 37 °C), loaded into NC-Slides^™^ A8, and analyzed using an image cytometer.

### 2.8. Statistical Analysis

Mean values of at least three separate experiments conducted in triplicate ± standard deviation of the mean (SD) were calculated. Differences between groups were evaluated using one-way ANOVA followed by Tukey’s and Dunnett’s post hoc comparison tests. A statistically significant difference was found at a p-value lower than 0.05. Statistical analysis was performed using GraphPad Prism 8.0 (GraphPad Software, San Diego, CA, USA).

## 3. Results

### 3.1. The Influence of NEO, Doxorubicin, Etoposide, and Irinotecan on the Cell Viability in Single and Combined Treatments

In order to assess the effect of NEO on the activity of doxorubicin, etoposide, or irinotecan in anaplastic astrocytoma SW1783 cells, cell viability was examined via the WST-1 test. Firstly, the treatment with single substances was conducted (Figure 1). Cells were incubated for 48 h at concentrations of NEO spanning from 1 μM to 75 μM, and chemotherapeutics ranging from 1 μM to 50 μM. The results indicated that all tested substances affect the viability of SW1783 cells in a concentration-dependent manner. The lowest concentrations at which a significant difference was observed were 10 μM of NEO (*p* < 0.05), etoposide (*p* < 0.001), and irinotecan (*p* < 0.05); and 1 μM of doxorubicin (*p* < 0.05). The percentages of viable cells in these treatments were: ca. 82%, ca. 52%, ca. 84%, and ca. 85% of the control, respectively. At higher concentrations of doxorubicin (10–50 μM), etoposide (50 μM), and irinotecan (50 μM), a severe decrease in viability of the treated cells was noticed, which was lower than approx. 32% of control.

In the next part of the experiment, the estimation of the viability of SW1783 cells incubated with the combination of NEO and a chemotherapeutic was conducted. Drug concentrations for further studies were selected based on the screening analysis of cell viability. They were the lowest concentrations causing a statistically significant difference, i.e., 1 μM of doxorubicin, 10 μM of etoposide, and 10 μM of irinotecan. For NEO, 25 μM was selected, as the difference in the cell viability between groups treated with 10 μM and 25 μM was small (ca. 3%). Moreover, 25 μM of NEO was utilized for combined treatment in our previous works [43,44]. Additionally, the highest concentration of NEO used in the earlier part (Figure 1A) was chosen, 75 μM.

The results revealed that in every group treated with the combination of NEO and a chemotherapeutic, there was a statistically significant reduction in cell viability when compared to the group treated with only a drug (Figure 2). The viability of cells incubated with the NEO–doxorubicin mixtures was estimated at approx. 18% for the group in which 25 μM of NEO was used and approx. 21% for a mix containing 75 μM of NEO. In cells treated with etoposide mixed with 25 μM of NEO, cell viability was at ca. 37% and with 75 μM of NEO at ca. 33% of control. The percentages of viable cells in NEO–irinotecan-treated groups were the highest among all cells incubated with the combination of the isoflavone and a chemotherapeutic. These were at approx. 68% for the group containing 25 μM of NEO and approx. 66% of control for 75 μM of NEO.

The results showed that there were no significant differences between the cell viability of groups co-treated with the chemotherapeutic and various concentrations of NEO. Therefore, for the cytometric analyses, the isoflavone concentration of 25 μM was used. Additionally, NEO–irinotecan treatment was excluded from further experiments since the cell viability of these groups did not differ significantly from the percentage of viable cells incubated with NEO (75 μM).

### 3.2. The Assessment of SW1783 Cell Growth Incubated with Doxorubicin or Etoposide Alone and in Combination with NEO

The cultures of SW1783 cells exposed to doxorubicin, etoposide, and their combination with NEO for 48 h were analyzed to examine the cell growth based on the total number of cells. It was estimated using the NucleoCounter^®^ NC-3000^™^ fluorescence image cytometer, as was reported in Section 2.4. The obtained results have shown that NEO (25 μM) when combined with etoposide (10 μM) does not affect the total number of SW1783 cells (Figure 3). In turn, there was a slight decrease (by approx. 12%) in the cell population treated with NEO–doxorubicin (25 μM + 1 μM) mix when compared to doxorubicin (1 μM) alone.

### 3.3. Combination of NEO and Etoposide Increases Apoptotic Subpopulation in SW1783 Cells

In order to quantitatively assess apoptosis in SW1783 cells incubated with the chemotherapeutics and their mixtures with NEO, an analysis involving Annexin V staining was performed. It allows for distinguishing cells in which phosphatidylserine externalization has occurred, which is one of the main characteristics of apoptosis [45]. The analysis showed that NEO (25 μM) increases the apoptotic subpopulation when combined with etoposide (10 μM) by ca. 24% compared to etoposide alone (Figure 4). This effect can be seen in cell subpopulations being in both early and late apoptosis. Between groups treated with NEO–doxorubicin mix (25 μM + 1 μM) and doxorubicin (1 μM) alone, no significant difference in the percentage of apoptotic subpopulations was present. In both of these groups, apoptosis was relatively high, approx. 95% and 94%, respectively.

### 3.4. Co-Treatment of NEO and Doxorubicin or Etoposide Prompts Changes in the Mitochondrial Membrane Potential of SW1783 Cells

Mitochondrial membrane potential (∆Ψm) is strictly related to the proper function of mitochondria. Its disruption usually occurs in stress conditions and is often associated with the process of apoptosis [46,47]. To investigate the effect of the NEO combination with doxorubicin and etoposide on the mitochondria, ∆Ψm was evaluated (Figure 5). When compared to the cells incubated with etoposide alone (ca. 10% of cells with low ∆Ψm), an increase in the percentage of cells with depolarized mitochondrial membrane was found in the NEO–etoposide-treated group and was estimated at ca. 58%. On the other hand, the co-treatment of NEO with doxorubicin at 1 μM caused an opposite effect on the ∆Ψm—in comparison to cells exposed to a single chemotherapeutic, the cell subpopulation characterized by low ∆Ψm was decreased in the group treated with the mixture (approx. 55% vs. 23%, respectively).

### 3.5. The Analysis of Cell Cycle in SW1783 Cells Treated with Doxorubicin or Etoposide Alone and in Combination with NEO

A cell-cycle analysis using NucleoCounter^®^ NC-3000^™^ fluorescence image cytometer was performed to examine the effect of NEO co-treatment with doxorubicin and etoposide on the distribution of cell-cycle phases (Figure 6). The data were recounted into a relative quantity ratio of the G_1_ and G_2_-M phases to the S phase. The results revealed that the combination of NEO with chemotherapeutics causes a reduction in the G_1_/S ratio. In the group treated with NEO–doxorubicin, the decrease in this ratio was almost 2-fold, and for NEO–etoposide, 1.6-fold. Moreover, they were significantly lower than the control (*p* < 0.001), in contrast to the groups incubated with a single chemotherapeutic. In the G_2_-M/S ratio, no differences between groups treated with only a drug and its combination with NEO were observed. However, a statistically significant reduction in comparison to the control was found in the NEO–etoposide sample, which was 1.6-fold.

## 4. Discussion

Despite the development of novel drugs, ineffective treatment remains a major problem of modern oncology. High-grade gliomas, such as anaplastic astrocytoma or glioblastoma, are characterized by poor prognosis regardless of radical therapy. Hence, there is a dire need for new therapeutical strategies. Anti-neoplastic agents used widely in clinical practice, such as topoisomerase inhibitors, are one of the subjects that are being investigated for their potential application in high-grade glioma chemotherapy [33,34,35,36,37]. However, their efficacy might be negligible when used in monotherapy. Targeting multiple pathways involved in the pathophysiology of cancer cells gives a greater chance of success; therefore, combination regimens are considered to be more potent [48].

NEO is a flavonoid (isoflavone) contained in *Psoralea corylifolia* (L.) that exerts anti-inflammatory [49], osteogenic [50], and anticancer [51] properties. In several in vitro studies, it was found to be effective in various neoplastic cell lines [44,51,52,53,54]. In our previous works, we aimed at studying the anticancer activity against the human glioblastoma U-87 MG cell line in relation to its potential chemosensitizing properties towards doxorubicin and etoposide [43,44]. In this paper, we examined if NEO can influence the in vitro activity of these chemotherapeutics in a different high-grade glioma cell line, human SW1783 anaplastic astrocytoma cells.

We found that NEO alone affects the viability of SW1783 cells in a dose-dependent manner (Figure 1A), to a similar extent as U-87 MG glioblastoma cells [44]. In turn, based on the results of the cell viability in groups treated with single chemotherapeutics (Figure 1B–D), (doxorubicin, etoposide, irinotecan) it seems that the SW1783 cell line is more susceptible to them than U-87 MG cells. Moreover, unlike the glioblastoma cells, the analysis of the viability of the anaplastic astrocytoma cells incubated in combined treatments revealed that the viability was significantly reduced in every group where NEO was present when compared to the drug alone (Figure 2). Further analysis of the cell count showed that NEO caused a decrease of proliferation only in the SW1783 cells co-treated with doxorubicin (1 μM) (Figure 3), which is consistent with our previous study where the inhibition of cell growth in the corresponding group of U-87 MG cells was also observed [43].

The main characteristic of neoplastic cells besides the uncontrollable growth is an ability to avoid programmed cell death [55]. For this reason, the induction of apoptosis is an important target of anticancer treatment [55,56]. Many of the drugs used in chemotherapy are aimed at activating apoptosis through the inhibition of DNA replication, such as topoisomerase inhibitors [57]. Herein, we found that the externalization of phosphatidylserine, a feature of the apoptotic cells, was caused by etoposide (10 μM) and doxorubicin (1 μM) in single treatment regimens in most of the population of SW1783 cells (Figure 4). Among the co-treated groups, NEO enhanced the pro-apoptotic effect of etoposide, but not doxorubicin. In turn, DNA fragmentation (sub-G_1_ phase) was not observed in any of the tested groups (Figure 6A), which may indicate an early stage of apoptosis. In our previous work [43], we obtained different results, which showed that NEO intensified apoptosis induced by etoposide and doxorubicin in U-87 MG glioblastoma cells. We hypothesize that the lesser effect of chemosensitization in SW1783 cells might be connected with the fact that, unlike the U-87 MG cell line, these cells are more prone to the action of single drugs.

The status of the mitochondria reflects the condition of the cell [47]. These organelles are involved in numerous crucial cellular metabolic processes and also regulate the process of apoptosis [46,47]. One of the first events that lead to the release of the pro-apoptotic proteins such as cytochrome C and the activation of the caspase cascade is the permeabilization of the mitochondrial membrane, which is accompanied by its potential (∆Ψm) disruption [46]. In the experiment, we examined ∆Ψm to determine if NEO influences the status of mitochondria when combined with etoposide or doxorubicin (Figure 5). In the first-mentioned group, we observed a strong increase in the subpopulation with the disrupted ∆Ψm, which might be assumed as an intensification of apoptosis through the intrinsic pathway. An opposite result was found in the NEO–doxorubicin group, where a decrease in the percentage of cells with low ∆Ψm was noticed, despite relatively high apoptosis (Figure 4). In the previous work conducted on U-87 MG cells [43], we obtained similar results in all groups where NEO was present, which indicates that SW1783 cells incubated with the mix of NEO and etoposide are an exception. This might be due to the differences between the cell lines such as an expression of specific proteins, which needs to be explored further.

The cell cycle is a multi-stage process in which cells undergo division. It comprises the G_1_ phase, where the cell grows, the S phase, in which DNA is being synthesized, the G_2_ phase, where the cell prepares for division, and finally, mitosis (M). There is also the G_0_ phase, which is referred to as the resting phase [58]. Dysregulation of the cell cycle often results in abnormal proliferation, which takes place in neoplastic cells [58,59]. Thus, cell-cycle inhibitors have great potential in anticancer therapy [59]. Topoisomerase inhibitors are considered to have phase-specific action, which causes an arrest in the S or G_2_/M phase [60]. In our study, we demonstrated that in SW1783 cells only doxorubicin has this effect (Figure 6). In turn, we found that the co-treatment of chemotherapeutics with NEO prompts changes in the cell-cycle distribution, which was characterized by the decrease in the G_0_-G_1_/S ratio when compared to the corresponding drug alone. Due to the fact that no changes in the G_2_-M/S ratio have been spotted, this may indicate an increase in both the S and G_2_/M phases, perhaps at the expense of the G_0_/G_1_ phases. It may also denote that the observed decrease in cell number in the doxorubicin-NEO group could be caused by the inhibition of the cell cycle in the G_2_ and/or M phases. Hence, this may mean that NEO intensifies the activity of etoposide and doxorubicin connected with the phase-specific arrest. This is in line with the results from our previous work, where NEO also caused a similar effect in U-87 MG cells [43].

To sum up, in this paper we assessed the potential chemosensitizing action of NEO towards doxorubicin and etoposide in SW1783 anaplastic astrocytoma cells. In reference to our previous works, this allowed us to conclude that this activity differs between cell lines of high-grade gliomas. These dissimilarities concern the effects on cell viability, apoptosis enhancement, and the status of mitochondria. In contrast, we found that in both cell lines NEO had corresponding effects on the cell count and the cell cycle.

## 5. Conclusions

In this paper, the potential chemosensitizing properties of NEO were evaluated. In the study conducted on human SW1783 anaplastic astrocytoma cells, it has been demonstrated that the co-treatment of NEO with etoposide results in the enhancement of apoptosis and an increase in the low mitochondrial membrane potential subpopulation. No effect on the apoptosis-inducing activity of doxorubicin was found. Nonetheless, NEO lowered cell viability and prompted changes in the cell cycle in all combined treatments.

## Figures and Tables

**Figure 1 cells-12-00593-f001:**
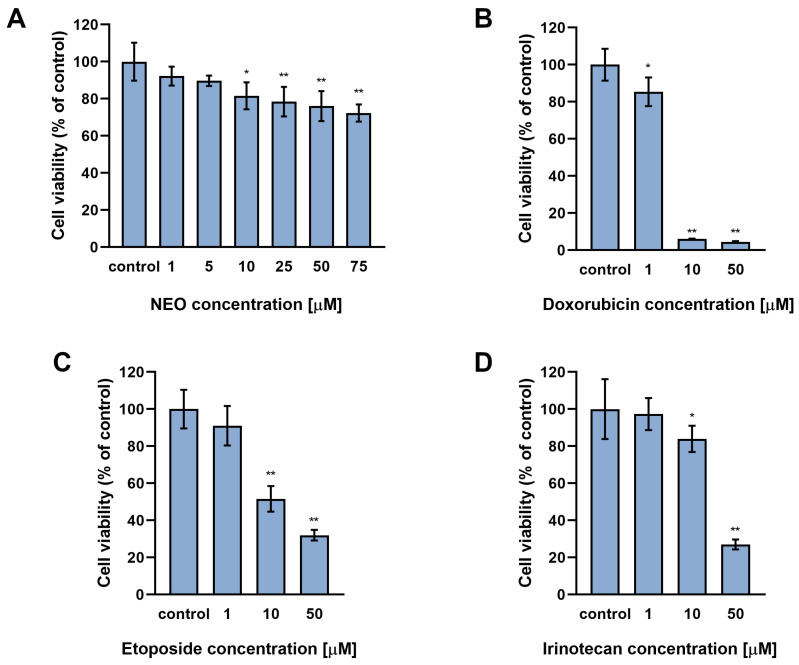
The impact of NEO (**A**), doxorubicin (**B**), etoposide (**C**), and irinotecan (**D**) on the viability of SW1783 cells. Cells were incubated with substances at indicated concentrations for 48 h and examined by the WST-1 test. The results are presented as a percentage of the controls. * *p* < 0.05, ** *p* < 0.001 vs. control.

**Figure 2 cells-12-00593-f002:**
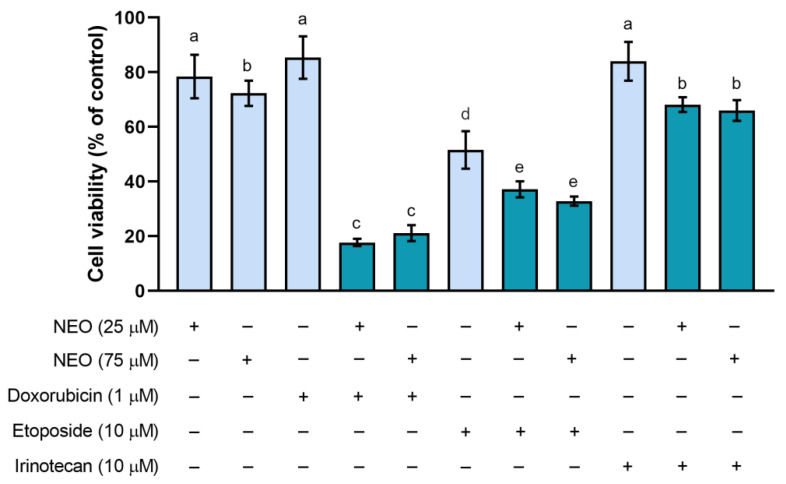
SW1783 cells viability comparison among groups treated for 48 h with only NEO (25 μM and 75 μM), a drug alone: doxorubicin (1 μM), etoposide (10 μM,) and irinotecan (10 μM), and NEO–drug mixtures (bars indicated in a darker color). Mean values ± SD from three independent experiments are presented. Means not sharing a common superscript differ significantly at *p* < 0.05.

**Figure 3 cells-12-00593-f003:**
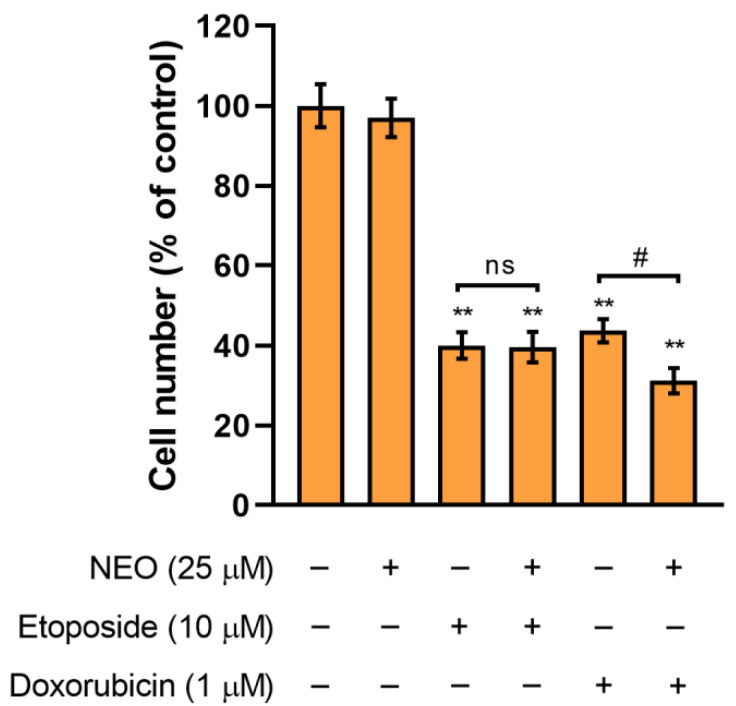
The effect of the combined treatment of NEO (25 μM) and doxorubicin (1 μM) or etoposide (10 μM) on the growth of SW1783 cells. Data are presented as percentages of the control. ** *p* < 0.001 vs. control; # *p* < 0.05 for comparisons between groups; ns—non-significant difference.

**Figure 4 cells-12-00593-f004:**
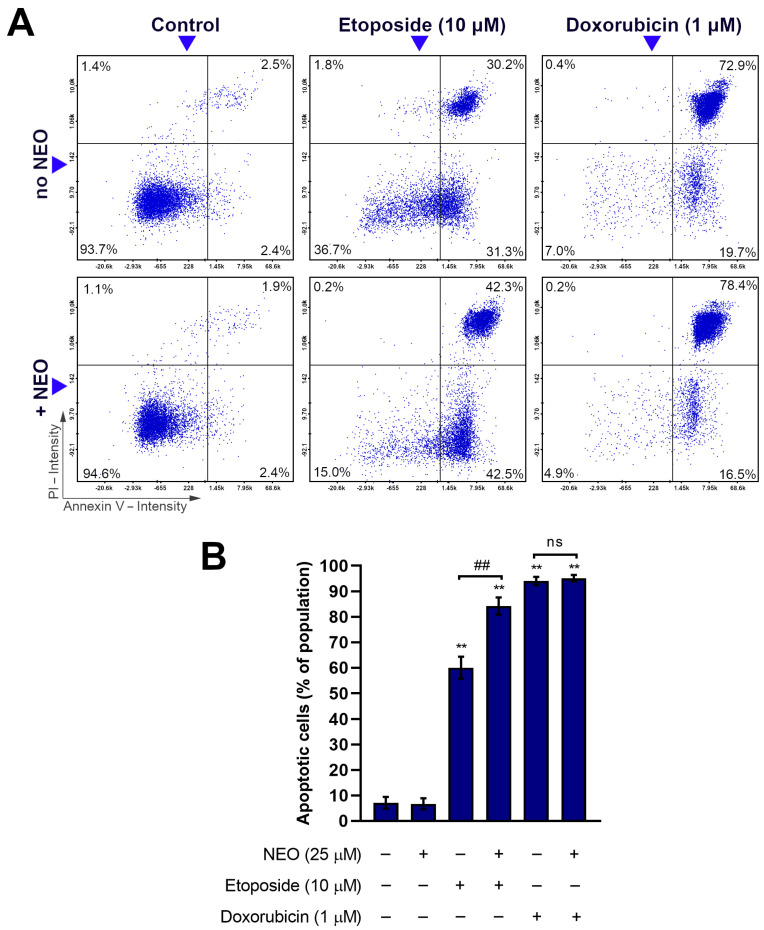
The detection of apoptotic subpopulation by Annexin V assay in SW1783 cells treated for 48 h with doxorubicin (1 μM) or etoposide (10 μM) alone and their mixes with NEO (25 μM). (**A**) Scatter plots showing cell population divided into four quadrants: lower left—healthy cells (Annexin V-negative/PI-negative), lower right—early apoptotic cells (Annexin V-positive/PI-negative), upper right—late apoptotic cells (Annexin V-positive/PI-positive), and upper left—non-apoptotic dead cells (Annexin V-negative/PI-positive). (**B**) Mean values ± SD of the percentage of apoptotic cells (Annexin V-positive) from three independent experiments are displayed in the bar graph. ** *p* < 0.001 vs. control; ## *p* < 0.001 for comparisons between groups; ns—non-significant difference.

**Figure 5 cells-12-00593-f005:**
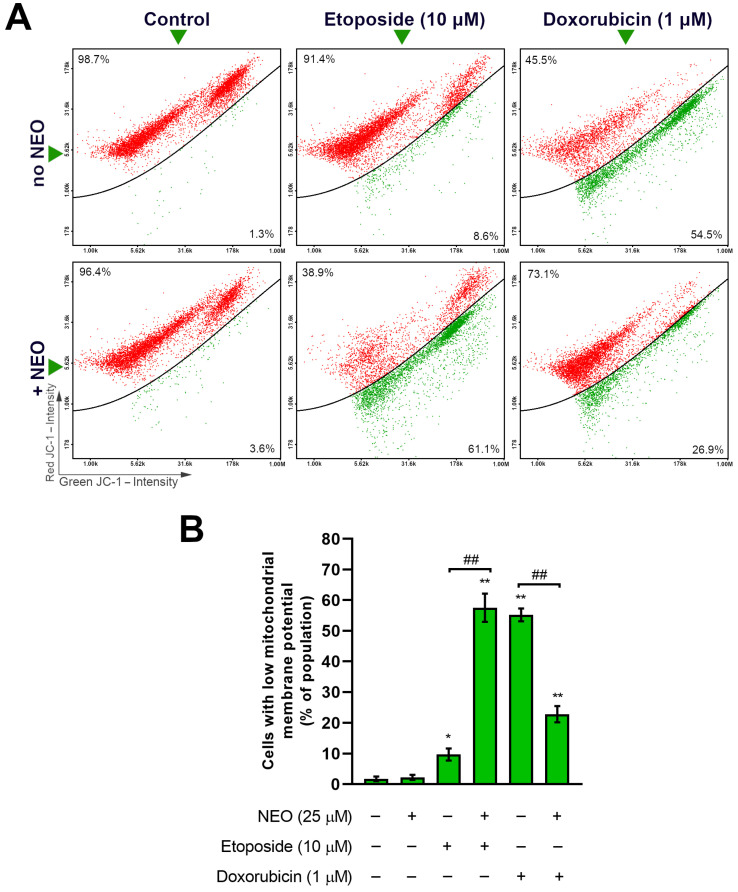
The assessment of the membrane mitochondrial potential in SW1783 cells incubated for 48 h with doxorubicin (1 μM) or etoposide (10 μM) alone and their mixes with NEO (25 μM). (**A**) Representative scatter plots showing cell population consisting of two subpopulations—cells with high mitochondrial membrane potential (red) and cells with low mitochondrial membrane potential (green). (**B**) Bar graph presenting mean values ± SD of the percentage of cells with decreased mitochondrial membrane potential from three independent experiments. * *p* < 0.05, ** *p* < 0.001 vs. control; ## *p* < 0.001 for comparisons between groups; ns—non-significant difference.

**Figure 6 cells-12-00593-f006:**
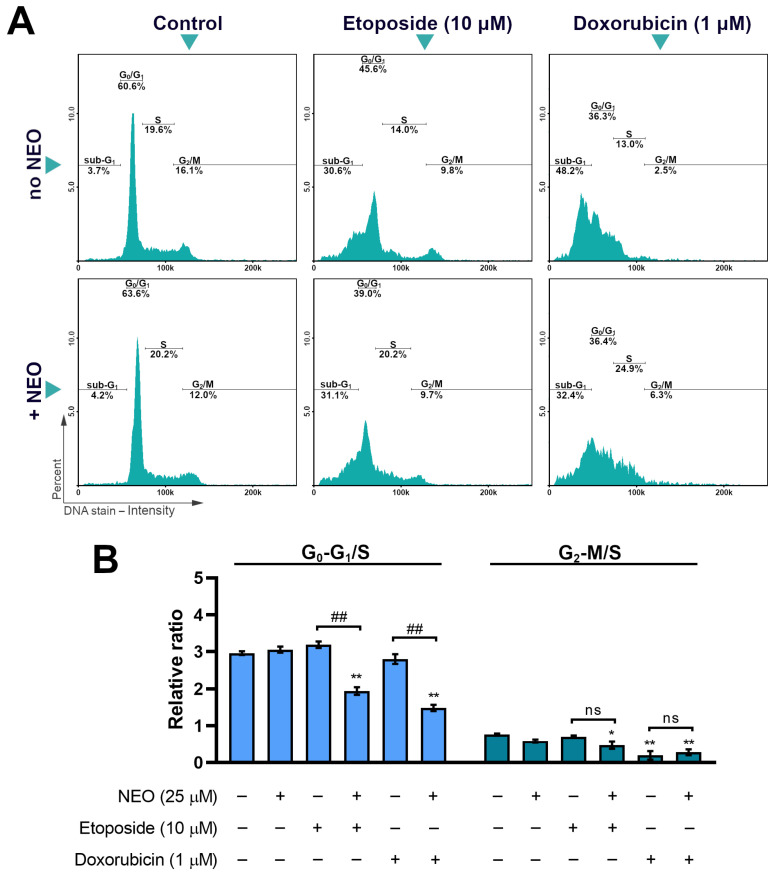
Changes in the cell cycle in SW1783 cells treated with doxorubicin (1 μM) or etoposide (10 μM) alone and their mixes with NEO (25 μM) for 48 h. (**A**) Representative histograms displaying the distribution of cell subpopulations in the cell-cycle phases. (**B**) Bar graph presenting results as a relative quantity ratio of G_0_-G_1_/S and G_2_-M/S. * *p* < 0.05, ** *p* < 0.001 vs. control; ## *p* < 0.001 for comparisons between groups; ns—non-significant difference.

## Data Availability

The data that support the findings of this study are available from the corresponding author upon reasonable request.

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
