# Peer review of "Evaluation of Possible Neobavaisoflavone Chemosensitizing Properties towards Doxorubicin and Etoposide in SW1783 Anaplastic Astrocytoma Cells"

_cells, 2023, doi:10.3390/cells12040593_

Round 1
Reviewer 1 Report
The paper entitled “Evaluation of Possible Neobavaisoflavone Chemosensitizing Properties towards Doxorubicin and Etoposide in SW1783 Anaplastic Astrocytoma Cells” found that combinational use of neobavaisoflavone could sensitize SW1783 cells to the treatment of doxorubicin and etoposide by inducing apoptosis and cell cycle arrest. However, the chemosensitizing properties of neobavaisoflavone have already been reported elsewhere, which makes this manuscript lack novelty. Moreover, some experiments are not rational enough and the conclusions are not well supported by the data. Therefore, the reviewer would recommend the manuscript rejection based on the following concerns:
1. Previous studies have already reported the chemosensitization of U-87 MG glioblastoma cells by neobavaisoflavone towards doxorubicin and etoposide (PMID: 35628432). Moreover, it has been reported that neobavaisoflavone sensitized apoptosis via the inhibition of metastasis in TRAIL-resistant human glioma U373MG cells (PMID: 24231449).
2. As the authors described in the “Introduction” part, the resistance to temozolomide caused by expression of MGMT often occurs in most cases of high-grade gliomas. Why the authors study the possible neobavaisoflavone chemosensitizing properties towards doxorubicin and etoposide, rather than temozolomide?
3. Figure 2 lacks statistics.
4. EdU incorporation assay is recommended to validate the effect of neobavaisoflavone on cell proliferation.
5. TUNEL and WB assays are suggested to validate the effect of neobavaisoflavone on the induction of apoptosis.
6.The authors are should investigate the potential mechanisms involving the chemosensitizing properties of neobavaisoflavone.
7. In vivo assays are needed to further validate the chemosensitizing properties of neobavaisoflavone.
Author Response
POINT-BY-POINT RESPONSE TO THE REVIEWER 1
We would like to thank the Reviewer for the assessment of our manuscript. We are grateful for the comments and valuable advices. We hope that the introduction of appropriate changes and the clarification of mentioned concerns will be satisfying.
Firstly, we would like to address the concern connected with the conclusions. We believed that the conclusions subsection should include mainly the summary of the obtained results. Being aware of the need to continue the research, we avoided unnecessary conclusions and over-interpretations.
Other addressed issues:
- “Previous studies have already reported the chemosensitization of U-87 MG glioblastoma cells by neobavaisoflavone towards doxorubicin and etoposide (PMID: 35628432). Moreover, it has been reported that neobavaisoflavone sensitized apoptosis via the inhibition of metastasis in TRAIL-resistant human glioma U373MG cells (PMID: 24231449).”
Answer:
In our previous work (PMID: 35628432), we evaluated the chemosensitizing properties of neobavaisoflavone in U-87 MG glioblastoma cells towards doxorubicin and etoposide, which are anticancer agents. We would like to emphasize that in the present study, we used a cell line derived from a different type of tumor – the cells that were used previously had been derived from WHO grade IV glioma, known as glioblastoma, while in our current manuscript WHO grade III glioma cell line was utilized, which is anaplastic astrocytoma. Despite the fact that both of these neoplasms are classified as high-grade gliomas, there are phenotypic and pathophysiological differences between them (PMIDs: 29200110, 24193082, 26948349). We believe that determining whether neobavaisoflavone exerts similar effects in different cell lines is worthful as it might not be universal. In this aspect, our findings presented in this manuscript shed more light on the properties of this isoflavone.
- “As the authors described in the “Introduction” part, the resistance to temozolomide caused by expression of MGMT often occurs in most cases of high-grade gliomas. Why the authors study the possible neobavaisoflavone chemosensitizing properties towards doxorubicin and etoposide, rather than temozolomide?”
Answer:
The anti-neoplastic agents, on which our research is focused, are topoisomerase inhibitors. It has been shown in several pre-clinical and clinical studies (examples, PMIDs: 19920819, 31689481, 30606517, 30639216) that they have a certain potential against gliomas. In one of our previous papers (PMID: 34361668) conducted on U‑87 MG cells and normal human astrocytes, we carried out a screening of several anticancer agents, such as doxorubicin, etoposide, and irinotecan, in order to check if neobavaisoflavone affects their activity against cells. We found this effect in co-treatments where doxorubicin or etoposide was present. In this paper, temozolomide was also studied, however, we found that neobavaisoflavone does not enhance its anti-glioma effect.
- “Figure 2 lacks statistics.”
Answer:
The statistics in the Figure 2 are present. They can be seen as superscripts above bars, which indicate statistical significant differences between mean values. As reported in the figure description, a mean labeled with a specific letter differs significantly at p<0.05 from another mean, which does not share the same letter. The intention of this format is to show multiple comparisons between each mean values. It is used in some research articles (examples, PMIDs: 34609608, 24288022, 19616452, 31873092, 33753826, 35851065, 33672156, 35273602, 28246309, 28701075, 27803291).
- “EdU incorporation assay is recommended to validate the effect of neobavaisoflavone on cell proliferation.”
Answer:
In our study, we assessed the number of cells and presented data as a percentage of the control, which was untreated cells. At the beginning of incubation, cell populations in each group were started by seeding an equal number of cells and were cultured in parallel under the same conditions. To estimate the quantity of populations, we used an image cytometer – detached cells were transferred into a cassette containing acridine orange. This dye stains every cell by binding to DNA, which is accompanied by green fluorescence. Each cell revealed in the microscopic image through the emission of fluorescence was counted by the image cytometer. In addition, the cell cycle profile of the tested groups was analyzed. Considering the obtained results, the Reviewer’s suggestion regarding the use of EdU incorporation assay is insightful. We are assured that this test will be used in the next step of our studies. This assay will allow us to evaluate in detail cell proliferation based on the ability to of the cell to replicate its genetic material. We are thankful for such a useful advice.
- “TUNEL and WB assays are suggested to validate the effect of neobavaisoflavone on the induction of apoptosis.”
Answer:
We are grateful for the Reviewer’s comment concerning the suggestion of carrying out TUNEL assay and Western Blot analyses. There are many tests that allow examining the induction of apoptosis. Each of them focuses on various characteristics of this process. In our manuscript, Annexin V assay was used to determine apoptosis in the cell populations. The main advantage of this test is its high sensitivity – Annexin V has a strong affinity towards externalized phosphatidylserine. Moreover, it can be detected at an early stage of apoptosis, because the enzymes responsible for the translocation of lipids within the cell membrane, the scramblases, are activated in the execution phase of apoptosis leading to a loss of lipid bilayer asymmetry (PMID: 23979019). The assay that was used in our study had been composed of not only fluorescent-labeled Annexin V but also propidium iodide, which penetrates cells where the membrane was compromised, i.e. dead cells, staining double-stranded nucleic acids. The benefit of this method is the fact that the use of both of these dyes allows the distinction of four subpopulations within cultured cells, as it was outlined in the Figure 4 description in the manuscript. The proposed by the Reviewer TUNEL assay is based on another characteristic of apoptotic cell death – DNA fragmentation, which takes place at the late stage of this process. The method relies on terminal deoxynucleotidyl transferase catalyzing a reaction, in which fluorescent-labeled deoxynucleotides are attached to the 3’-hydroxyl breaks of DNA. The subpopulation of cells with fragmented DNA can also be spotted during the analysis of the cell cycle based on the intercellular DNA content. It is characterized by having less than one equivalent of DNA, which is known as a sub-G1 subpopulation. This method is also utilized to assess DNA fragmentation, and therefore apoptosis (examples, PMIDs: 26434128, 20361030, 26148186, 29391428, 30941553). In the discussion section of our manuscript, we noted that no increase in the late apoptotic subpopulations with fragmented DNA (sub-G1 subpopulations) was found in groups treated with both neobavaisoflavone and the drug when compared to the chemotherapeutic alone (Figure 6A of the manuscript).
- “The authors are should investigate the potential mechanisms involving the chemosensitizing properties of neobavaisoflavone.”
Answer:
We understand the Reviewer’s remark regarding the potential mechanisms underlying properties of neobavaisoflavone. We understand that this issue is very important, but also very complex. This manuscript is a part of a larger project and we believe that assessment of the properties of neobavaisoflavone in different cell lines can contribute to finding out the chemosensitizing mechanism of neobavaisoflavone.
- “In vivo assays are needed to further validate the chemosensitizing properties of neobavaisoflavone.”
Answer:
The presented study is a part of the initial stage focused on in vitro research and we think that this subject has not yet been exhausted. We would like to emphasize that this model is consistent with the aims and scope of the journal, which prioritize experimental cytology. We believe that our paper might be an excellent basis for further research, including in vivo studies. Thus, we are certain that the Reviewer’s profound opinion will be taken into account in the future.
Reviewer 2 Report
General Comments:
The manuscript presents the effect of the flavonoid compound Neobavaisoflavone (NEO) as an enhancer of two anti-cancer drugs: etoposide and doxorubicin in the narrow context of the human SW1783 anaplastic astrocytoma cell line. Combining NEO to etoposide (ETO) or doxorubicin (DOX) caused the death of SW1783 cells at concentrations lower than when the drugs were used alone—in order words: NEO showed a synergestic effect. Four prelimary investigations using phenotypic assays revealed that the NEO+ETO and NEO+DOX most likely act using different pathways, but the exact mechanisms are not known. Specifically: (1) the drugs combination showed minimal to no effect on cells growth, (2) apoptosis induction was observed only with NEO+ETO, (3) the changes in mitochondrial membrane potential were significant, but opposite with each combination, and (4) the effect of both drug combos appear to take place during the early phases of the cells’ cycles.
This study on NEO’s synergistic effects is an extension on two precedents reports from the same authors with NEO alone (Molecules 2021), and with NEO+ETO and NEO+DOX in an very similar cells line: U-87 astrocytoma (Int. J. Mol. Sci. 2022). The only difference between the other 2022 paper and this manuscript is the cell line: the same assays and drugs combos were tested with human U-87 MG glioblastoma cells, whereas this paper now looks at human SW1783 anaplastic astrocytoma cells.
This the data is interesting, as the effects on SW1783 cells appears slightly more potent than on U-87 cells (but SW1783 cells are known to be more naturally sensitive).
The data are presented clearly and logically, but the novelty vs. the previous IJMS paper is not obvious. The article would greatly benefit from more mechanistic insight. While the conclusions are supported by the data, the discussion is limited to observational comments.
Significance: Given the fact that NEO is a natural product, and that the other drugs are already approved in human, there is potential for a very rapid translation into the clinic… that said, as recognized by the auhtors, flavonoids like NEO are likely already present in our food, so is it possible that a metastudy could support their hypothesis?
I support publication of the manuscript after the following changes.
Minor modifications:
• Cell cycles cyctometry analyses: explain why the “S” phase was selected as reference? this is especially dubious in the case of Doxorubicin (Fig. 6A), where the histogram data shows very poor resolution, i.e., there is no defined peak or trough! Should there not be an internal standard here? otherwise, this is a self-referencing relative ratio… these error bars are meaningless—the cell cycle is a continuum, and clearly, the drugs affect several phases; how were they calculated?
[94-95 / 158] solutions 7, 8, 15, and 16 are prepared in what solvent?
[106] what is the % FBS used?
[110 / 121] the drug solutions were prepared in what?
Grammar/syntax to be fixed for proper English or clarity: lines: 197-199, 221 (in overall), 340 (both used), 346 (unalike).
• Statements from lines 342-345 are unclear and must be rewritten or expanded.
• [376-380] The G0/G1 phases are presented as “decreased” when the co-treatment is used… but this is a relative ratio… why is it not an increase of the S-phase? for the same reason as above, the low resolution of the cytometry data does not allow for a robust conclusion here.
Author Response
POINT-BY-POINT RESPONSE TO THE REVIEWER 2
We would like to thank the Reviewer for the assessment of our manuscript. We are grateful for the comments and valuable advices that would help improving the work. According to the suggestions, we introduced following changes and corrections to the paper. All edited fragments of the manuscript are to be found in bold.
Addressed issues:
- “Cell cycles cytometry analyses: explain why the “S” phase was selected as reference? this is especially dubious in the case of Doxorubicin (Fig. 6A), where the histogram data shows very poor resolution, i.e., there is no defined peak or trough! Should there not be an internal standard here? otherwise, this is a self-referencing relative ratio… these error bars are meaningless—the cell cycle is a continuum, and clearly, the drugs affect several phases; how were they calculated?”
Answer:
In the analysis based on the intercellular DNA content, such as the one that was used in our study, the sub-G1 subpopulation is also revealed beside the “standard” phases of the cell cycle, which are late-apoptotic cells with fragmented DNA. Due to the large percentage of this subpopulation, the results of the cell cycle phases distribution can be distorted and the conclusions may be flawed. Therefore, conversion to ratios allows one to eliminate this issue.
During the conversion to ratios, the S phase of the cell cycle was chosen as a consequent due to the fact that it is a common element connecting the G1 and G2 phases. This way of presenting the results of cell cycle analysis can be found in some research articles (examples, PMIDs: 23744359, 20041160, 20433742). In our manuscript, the G0-G1/S and G2-M/S ratios were calculated based on the results of one experiment, i.e. percentages of subpopulations that have been in the respective phase were used. Mean values and standard deviations were calculated based on the ratios of all repetitions of experiments.
Regarding the Reviewer’s remark concerning indistinct peaks in some histograms: all markers were applied based on the phases distribution of the control. In each experiment, in every tested group and the control cells were seeded in equal amounts and cultured in parallel under the same conditions. The analysis of the cell cycle was performed on the same number of cells.
- “[94-95 / 158] solutions 7, 8, 15, and 16 are prepared in what solvent?
[106] what is the % FBS used?
[110 / 121] the drug solutions were prepared in what?”
Answer:
According to the Reviewer’s suggestion, following changes were introduced to the manuscript:
2.1. Reagents and Chemicals:
“...Solution 7 (200 μg/mL JC-1 in DMSO), Solution 8 (1 μg/mL DAPI in PBS), Solution 15 (500 μg/mL Hoechst 33342, aqueous), Solution 16 (500 μg/mL propidium iodide, aqueous) were retrieved...”
2.2. Cell culture:
“...Cells were cultured in DMEM supplemented with FBS (final concentration 10%)...”
“...Afterwards, the medium was removed and the drug solutions, prepared in DMEM, were added...”
- “Grammar/syntax to be fixed for proper English or clarity: lines: 197-199, 221 (in overall), 340 (both used), 346 (unalike).”
Answer:
Taking into account the Reviewer’s remark, we modified mentioned sentences:
3.1. The Influence of NEO, Doxorubicin, Etoposide, and Irinotecan on the Cell Viability in Single and Combined Treatments:
“In the next part of the experiment, the estimation of the viability of SW1783 cells incubated with the combination of NEO and a chemotherapeutic was conducted. Drug concentrations for further studies were selected basing on the screening analysis of cell viability. They were the lowest concentrations causing a statistically significant difference, i.e.: 1 μM of doxorubicin...”
“The results showed that there were no significant differences between the cell viability of groups co-treated with the chemotherapeutic and various concentrations of NEO...”
- Discussion:
“Herein, we found that the externalization of phosphatidylserine, a feature of the apoptotic cells, was caused by etoposide (10 μM) and doxorubicin (1 μM) in single treatment regimens in most of the population of SW1783 cells...”
“...we obtained different results, which showed that NEO intensifies apoptosis induced by etoposide and doxorubicin in U-87 MG glioblastoma cells”
- “Statements from lines 342-345 are unclear and must be rewritten or expanded.”
Answer:
According to the Reviewer’s comment, changes were introduced to the manuscript and are to be found in bold.
- Discussion:
“Among the co-treated groups, NEO enhanced the pro-apoptotic effect of etoposide, but not doxorubicin. In turn, DNA fragmentation (sub-G1 phase) was not observed in any of the tested groups (Figure 6A), which may indicate an early stage of apoptosis. In our past work...”
- “[376-380] The G0/G1 phases are presented as “decreased” when the co-treatment is used… but this is a relative ratio… why is it not an increase of the S-phase? for the same reason as above, the low resolution of the cytometry data does not allow for a robust conclusion here.”
Answer:
We agree with the Reviewer’s remark concerning the interpretation of the relative ratios. According to this comment, we modified the Discussion section to emphasize the S phase increase rather than G0/G1 phases reduction. We also indicated a possible connection between cell proliferation and the changes in the cell cycle. The changes to the manuscript are to be found in bold.
“Due to the fact that no changes in the G2-M/S ratio were spotted, this may indicate an increase in both the S and G2/M phases, perhaps at the expense of G0/G1 phases. It may also denote that the observed decrease in cell number in doxorubicin-NEO group, could be caused by the inhibition of cell cycle in the G2 and/or M phases. Hence, this may mean that NEO intensifies...”
Reviewer 3 Report
In this paper Maszczyk et.al. describe a study of co-administration of some antineoplastic drugs with flavonoid neobavaisoflavone. The latter is known to induce apoptosis in some tumor cell lines. Authors have found that the combination of NEO with doxorubicin or etoposide improve the cytotoxicity. A grate effort was done to clarify the mechanism of such a activity although without success.
The principal drawback of this paper is the choice to use doxorubicin o etoposide for treating CNS tumor cell lines. The major problem for such an application is a very low bioavailability of chemotherapic drugs in CNS (as author states in introduction section). And the combination with NEO could not influence the bioavailability in any way. So, the clinical application of such a drug combination remains impossible.
Apart from that the manuscript is well written and could be of some general interest and I suggest to accept it in a present form.
Author Response
POINT-BY-POINT RESPONSE TO THE REVIEWER 3
We would like to thank the Reviewer for the assessment of our manuscript. We are grateful for the valuable comments.
We agree with the Reviewer’s comment – poor distribution of the drugs into the central nervous system could be a serious therapeutic problem. However, modern therapeutic delivery systems may improve efficacy of the regimens based on the drugs with low bioavailability. In the presented manuscript, our aim was to evaluate the effectiveness of the tested substances in the in vitro model. We believe that the presented results set the direction for further studies on gliomas, and have scientific and substantive value.
Round 2
Reviewer 1 Report
Acceptable.